# Anemia among Adolescent Girls in West Java, Indonesia: Related Factors and Consequences on the Quality of Life

**DOI:** 10.3390/nu14183777

**Published:** 2022-09-13

**Authors:** Puspa Sari, Dewi Marhaeni Diah Herawati, Meita Dhamayanti, Dany Hilmanto

**Affiliations:** 1Doctoral Study Program, Faculty of Medicine, Universitas Padjadjaran, Bandung 45363, West Java, Indonesia; 2Department of Public Health, Faculty of Medicine, Universitas Padjadjaran, Bandung 45363, West Java, Indonesia; 3Department of Child Health, Hasan Sadikin Hospital, Faculty of Medicine, Universitas Padjadjaran, Bandung 45363, West Java, Indonesia

**Keywords:** anemia, adolescent girls, iron consume, menstruation, anthropometric measure, quality of life

## Abstract

Anemia in adolescent girls is still a problem in Indonesia. The impact of anemia is quite significant for adolescent health. This study aims to analyze related factors of anemia among adolescent girls, and the effect of anemia on the quality of life. The study was conducted in the Soreang District, West Java, Indonesia. This cross-sectional study involved 286 female students (15–19 years). A 24-h recall questionnaire was used to collect the nutrient intake. We use the WHOQOL-BREF to analyze the quality of life. The study assessed height, weight, body mass index (BMI), mid-upper arm circumference (MUAC), and a capillary blood sample to determine hemoglobin levels. Bivariate and multiple logistic regression tests were measured to find the factors most influencing anemia. The prevalence of anemia was 14.3%. Related factors of anemia in this study were: duration of blood show per menses, iron consumption, weight, height, and MUAC. From bivariate analyses, anemia influenced the social relationships domain with *p* < 0.05. Multivariate logistic regression showed that the most influencing factors for anemia were MUAC and duration of blood per menses. Anemia impacted the social relationships domain. In this study, the two main factors that affected anemia were MUAC and duration of blood each menstrual cycle.

## 1. Introduction

According to World Health Organization (WHO), adolescents are individuals aged 10–19 years [1]. Adolescence is a golden period with optimal growth and development. During this period, adolescents have essential physical and psychological needs. Moreover, if there is a nutritional deficiency, one of the problems is anemia. Anemia is a global public health problem affecting adolescent girls, women of childbearing age, pregnant women, and children in developed and developing countries [2,3,4,5,6,7]. Indonesia is one of developing country that still has stunting and anemia problems [8,9]. Anemia in Indonesia among women of reproductive age (15–49 years) has increased from 21.6% in 2018 to 22.3% in 2019 [9,10,11]. According to national data, prevalence of anemia that occurs in rural areas in Indonesia is greater than in urban areas [12]. Soreang District is a rural area and part of the Bandung District, West Java Province, where data on anemia and its causes are not yet available.

Anemia is a condition of low red blood cells (RBCs) or hemoglobin. Iron deficiency is the most common cause. Although iron deficiency causes a decrease in hemoglobin and RBCs production, which in turn lowers hemoglobin concentration and hematocrit, there are many other causes of anemia that do not involve iron [3,13]. Based on hemoglobin concentration, anemia is classified as mild, moderate, and severe [3]. Cut-off values for hemoglobin concentration of non-pregnant woman are: non-anemic; mild; moderate and severe is ≥12 g/dL; 11.0–11.9 g/dL; 8.0–10.9 g/dL; <8 g/dL, respectively [3]. According to WHO, due to the public health problem, anemia in a population is identified according to population prevalence as: not a public health problem (≤4.9%); mild (5.0–19.9%); moderate (20.0–39.9%); or severe (≥40.0%) [3].

Sickle cell anemia (SCA) and IDA have almost the same sign. Sickle cell anemia (SCA) is the most significant hemoglobinopathy in the world. Geographical regions with terrible socioeconomic situations are where SCA is most prevalent. Sickle cell anemia is most frequently found in Sub-Saharan Africa, the South Mediterranean regions, as well as in Middle Eastern and Indian ethnicities [14,15]. Epidemiological research has demonstrated that this condition has evolved naturally in tropical regions where malaria is prevalent [16]. Since the research area is not a malaria-endemic region, we did not conduct laboratory tests to determine whether or not SCA was prevalent. Low MCV and MCH can be signs of SCA, but other tests must be conducted. To exclude a diagnosis of SCA, we conducted an anamnesis and physical exam.

Anemia is part of malnutrition problems, with determinants: growth and development process, physiological, sex, age, and race, also associated with infection such as helminth infections, schistosomiasis, malaria, human immunodeficiency virus (HIV), tuberculosis, genetic disorders of hemoglobin, thalassemia, social behavioral, and environmental determinants [3,7]. In addition, the micronutrient deficiencies known to cause or contribute to anemia include vitamins A, B2, B6, B9, B12, C, D and E, and zinc, each acting through different mechanisms [3]. Nutrition consumed by adolescents determines nutritional status, in this case body mass index (BMI). Several studies have analyzed the relationship between anemia and BMI also mid-upper arm circumference (MUAC). [9,17,18,19] In the study reported by Nainggolan et al., in Indonesia, women overweight and obese were less likely to develop anemia than those with a normal BMI, regardless of their MUAC score [9]. Anemia is not only related to anthropometric measurements but also to parents’ education and income, as well as menstruation [20].

The lack of hemoglobin resulting from anemia limits blood oxygen transport, resulting in reduced physical and mental capacity, along with other health risks [3]. Nutrient intake correlates with the QoL [12]. Another study revealed that anemia was associated with lower quality of life in adolescents [21]. According to the WHO, Quality of life (QoL) is an individual’s perception of one’s position in everyday life, according to the culture and the value system in which they live and with their goals, hopes, desires, and concerns [22]. Reducing anemia among women of reproductive age is an important factor in the improvement of women’s health, children’s health, school performance, women’s work productivity, healthier pregnancy outcomes and intergenerational benefits for good health, economic and community development [3,4,6].

Multiple interventions are needed to address iron deficiency and anemia among women of reproductive age, including adolescent girls, pregnant women and postpartum women. Programs to prevent nutritional anemia usually focus on the provision of iron, folate, vitamin A, zinc and other micronutrients through different interventions, including supplementation, fortification and improvement of dietary diversity and food security (increased diversity of agricultural production, nutrition education, microfinance, women’s empowerment, targeted food distribution), as well as agricultural practices [3]. The WHO has recommended various programs, such as iron supplementation, food fortification, health education, and parasitic infection control, to reduce the prevalence of anemia [23]. Efforts to reduce the problem of anemia are one of the World Health Assembly’s Global Nutrition Targets for 2025 and the Sustainable Development Goals (SDGs), along with reducing stunting, wasting, and being overweight. Though some progress in lowering anemia has been achieved, it has not shown an optimal effect on anemia in women of reproductive age. One third of all women of reproductive age have anemia [24].

This study aims to analyse the factors that cause anemia and the effect of anemia on the quality of life in adolescent girls so that the prevalence of anemia can be prevented from an early age, starting from adolescence.

## 2. Materials and Methods

### 2.1. Study Design and Participants

This study was cross-sectional and conducted in Soreang, Bandung District, located 18 km south from Bandung City, which is the urban center. This area is a rural area, with the majority of the population being farmers and gardeners. This research was conducted in Soreang because, based on a preliminary study, it turned out that some adolescent girls did not know about anemia and the prevention. Based on data from the head of the health office, the proportion of those with anemia in the Bandung district area is 12.9% [25]. In contrast, for the Soreang area, there is no data on rate of Anemia that occurring in adolescent girls. Sample size was calculated with a statistical power analysis program (G*Power 3.1.9.7), with X^2^ test, 95% confidence level, 5% margin of error, the minimal sample was 220, the final sample was 286 adolescent girls. The consideration of choosingfemale students is to prepare them for marriage and healthy pregnancy because, in rural areas, adolescent girls marry after finishing high school. Adolescent girls who suffer from anemia have the potential risk of having a stunted child. Efforts to prevent anemia must be carried out early on. Therefore, this study took female students as research participants. Female students (15–19 years) from 16 schools in the Soreang sub-district were included in the study. Participants with a history of blood transfusion, currently being treated for anemia, worms, and thalassemia, were excluded.

Following the Declaration of Helsinki, this research has been approved by the Faculty of Medicine Ethics Committee Universitas Padjadjaran with the number 018/UN6.KEP/EC/2022. This study obtained the consent of by informing them about the research method.In addition, the parents or legal guardians of the adolescent girl also approved the informed consent.

### 2.2. Data Collection

This research was conducted between April and July 2022 to investigate the determinant factors of anemia among adolescent girls in Soreang Districts. We collected sociodemographic data, history of menstruation, history of health, personal hygiene, and physical activity. The physical activity was measured according to the international physical activity questionnaire (IPAQ), which was translated into the Indonesian version and has has been tested for validity and reliability [26]. A 24-h recall-structured questionnaire was used to analyze macronutrient and micronutrient intake. WHOQOL BREFF was used to analyses adolescent girls’ quality of life. Anthropometrics were measured to determine weight, height, BMI, and MUAC.

Data on macronutrient and micronutrient intake through a 24-h recall questionnaire were collected 30 minutes before blood sampling. The nutrition intake was taken three times, at the time of the initial data collection, 1.5 months after the initial data collection, and 1.5 months after the second data collection. In the 24-h recall questionnaire, there are also questions about dietary supplements that adolescent girls routinely consume.

#### 2.2.1. A 24-h Recall Questionnaire

The 24-h recall questionnaire was used to analyse adolescent nutrient intake. A 24-h dietary recall is a structured interview designed to obtain comprehensive information about all foods and drinks (as well as potential nutritional supplements) consumed by the respondent for the previous 24 h. In addition, it is among the simplest and most widely used ways to perform a food survey. According to studies, when used correctly, the 24-h recall method provides accurate information about the quantity and quality of food consumed. Nutri-Survey software was used to analyze energy, the macronutrient intake, namely: carbohydrate, protein, and fat and also micronutrient intake, namely: vitamin A, vitamin B1, vitamin B2, vitamin B6, vitamin C, vitamin E, calcium, folic acid, iron and zinc.

#### 2.2.2. WHOQOL BREF Questionnaire

The WHOQoL-BREF was used to determine the QoL of participants. This instrument categorizes QoL into four domains. WHOQOL BREF is scored in four domains, namely: physical health (domain 1), psychological (domain 2), social relationships (domain 3), and environment (domain 4) [13]. The average score of the items in each domain is used to calculate the domain’s value. The domain values are on a scale of 0–100 [22].

#### 2.2.3. Anthropometry Measurements

Weight was measured by OMRON digital weight scale to the nearest 0.1 kg. Height was measured with a Secca-stadiometer to the nearest 0.1 cm. The body mass index (BMI) was determined by calculating weight (kg)/height (m). Participants were classified as underweight, normal, and overweight, based on Indonesia BMI threshold from ministry of health [27]. Figure 1, show the measurement of mid-upper arm circumference (MUAC). MUAC of the dominant arm was measured by the mid-point between the tip of the shoulder and elbow [28].

#### 2.2.4. Laboratory Analyses

We analyzed venous blood samples for complete blood count and hemoglobin levels detected by Sysmex KX-21. Based on WHO classifications, hemoglobin levels below 8 g/dL were classified as severe anemia, 8–10.99 g/dL as moderate anemia, and 11–11.9 g/dL as mild anemia. Hb of 12 g/dL and above were classified as normal in adolescent girls [29]. Hematodrit (HCT), mean corpuscular volume (MCV), mean corpuscular hemoglobin (MCH), and mean corpuscular hemoglobin concentration (MCHC) can also be used as a measure to diagnose anemia, at the same time.

Blood sampling was carried out for one day for one school, where each school consisted of 5 to 29 participants. The time for taking the blood was 16 days, according to the number of schools. The time required for one school blood collection is about 2–3 h. Blood samples were taken immediately, by laboratory personnel, in each school, then sent to the nearest laboratory.

### 2.3. Statistical Analysis

Data were checked, cleaned, and coded using the IBM SPSS version 27 statistical software for windows. Descriptive statistic was used to participants’ characteristics. Bivariate logistic regression analyses were then performed to look for correlations between each independent variable and the outcome variable (anemia). All variables with *p*-value < 0.5 in the bivariate analysis were entered into a multivariable logistic regression analysis.

## 3. Results

### 3.1. Participants Characteristics

The characteristics of the participants are shown in Table 1. This study involved 286 adolescent girls aged 15–19 years. The participants were female students from 16 high schools in Soreang District. Adolescent girls who were not anemic were 245, while there were 41 anemic. There was no significant difference in parent education and occupation, living condition, siblings, monthly family income, menarche, sanitary pad usage per day, and dietary habits. In addition, we found that height for age (z-score) and BMI status between anemic and non-anemic have not significantly difference. Due to hemoglobin level, anemia in this study was in the moderate classification.

### 3.2. Nutrient Intake and Quality of Life

Table 2 shows from the median value, participants in this study had macronutrient and micronutrient intakes that were more than the reference nutrient composition of Indonesia. Nevertheless, from the minimum score, macronutrient and micronutrient intakes of participants were below the reference nutrient composition of Indonesia. Adolescents’ quality of life who were anemic and non-anemic from the physical health, psychological, and environmental domains showed no significant differences. Moreover, in social relationships, there were significant differences between anemic and non-anemic adolescents with *p* < 0.05.

### 3.3. Anthropometric and Hematology

Table 3 shows the significant difference in weight, height, White blood cells (WBC), Hemoglobin, HCT, MCV, MCH, and MCHC with *p* < 0.05.

### 3.4. Multivariate Analysis

The value of the difference test with *p* < 0.25, was entered into the variables for the logistic regression test, namely: duration of blood show per menses, iron consumed, weight, height, and MUAC. Table 4 shows the most related factors with anemia amongst adolescent girls in the Soreang District was the duration of blood show per menses and mid-upper arm circumference (MUAC).

## 4. Discussion

Anemia has an impact on adolescents’ health, performance, and productivity. It also leads to healthier pregnancy outcomes and benefits future generations’ health [3]. According to the WHO, anemia in our study was a mild public health problem. In addition, due to hemoglobin level, the anemia classification was moderate anemia. Based on bivariate analysis, we observed significant differences between anemic and nonanemic participants: duration of blood show per menses, weight, height, WBC, Hb, HCT, MCV, and MCHC with each *p* < 0.05. Our study also found that anemia was associated with quality of life in social relationships domain, with *p* = 0.024. After we performed a bivariate analysis, we performed a logistic regression analysis to determine the factors that most influence anemia. In this case, we included variables with a *p*-value of 0.25 in the bivariate analysis: duration of blood per menses, weight, height, MUAC, iron consumption, WBC, Hb, HCT, MCV, MCH, and MCHC. However, we did not include WBC, Hb, HCT, MCV, MCH, and MCHC because, theoretically, low levels of these blood components are indicators of hemoglobin counts [30]. The logistic regression results that most affected anemia was blood duration per menses and MUAC.

In terms of nutritional intake, there is no significant difference between anemic and non-anemic, but this should be highlighted because anemia is part of a nutritional problem. We found that in the anemic group, adolescents snacked more frequently (Table 1). This result is in line with national data, which states that adolescents have snack habits. The percentage of unhealthy food and drink consumption habits in the school-age population in 2018 is quite large, including sweet and fatty foods [8]. In line with this, previous studies found that adolescents consume more foods that contain carbohydrates and sweet foods, including chocolate [9]. The consumption of unhealthy snacks shows an unbalanced dietary intake (Table 2). The table indicates that some adolescents have intakes exceeding the national recommended daily allowance (RDA), but some have dietary intakes below the recommended ones. This impacts nutritional status where some adolescents are underweight, but some are overweight (Table 1). This includes a double burden of disease, as stated by WHO. Although malnourished children have gotten attention, the double burden of malnutrition (overweight and malnutrition) still occurs in Indonesia [4,10,11]. The three countries with the most children who are wasted are almost the same ones—India (25.5 million) and Nigeria (3.4 million) but also Indonesia (3.3 million). Countries that have more than a million children overweight include China, Indonesia, India, Egypt, US, Brazil and Pakistan [12]. However, undernutrition and overnutrition can trigger anemia, especially iron-deficiency anemia [31]. In addition, micronutrients deficiencies can cause anemia [32]. Health education to improve healthy dietary intake is essential for women to reduce the prevalence of anemia [9,33].

WHOQOL BREF scored in four domains, namely: physical health (domain 1), psychological (domain 2), social relationships (domain 3), and environment (domain 4) [22,34]. In this study, the quality-of-life domain that shows the difference is only domain 3, namely social relationships. Social relationship domains between anemic and non-anemic have a significant difference, with *p* = 0.024. The significant difference between the social relationship domains of anemic and non-anemic participants may be due to anemia symptoms, which cause weakness, tiredness, and lethargy, which will interfere with a person’s ability to socialize with others. In addition, nutritional deficiencies, including anemia, affect the growth and development of adolescents, which in turn affects their quality of life [35]. A study conducted by Seyed et al. found that anemia not only affects different aspects of their quality of life (QOl) but also the quality of life of their parents [36]. The significant difference in quality of life between anemic and non-anemic in this study is similar to a study conducted by Grondin et al. which revealed quality of life, significantly lower in the iron-deficient group [37]. On the other hand, several studies reveal a relationship between quality of life and anemia, which occurs in patients with anemia and other diseases such as cancer [37,38,39]. There can be different possible explanations for the difference in health-related quality of life between women with iron depletion and women who were iron-sufficient and did not differ significantly in self-perceived health, well-being, or fatigue [40]. Only a few studies have shown the association between anemia and the quality of life of adolescents who only have anemia. This data becomes essential for the following research and strength of this study, which analyses the relationship between anemia in anemic adolescents without other diseases and quality of life.

The logistic regression analysis showed a significant association of anemia with two variables, namely: duration of blood show per menses and MUAC. Our study found that duration of blood show per menses had significant correlation with anemia. This finding was in agreement with the finding which is reported by Hisa et al. that the risk of anemia is higher in women who have menstruation [30,41,42]. Moreover, that there was a significant association of Hb levels with MUAC, weight, and height, this was similar to a study conducted among in adolescent girls living in rural India. There is a relationship between the production of Hb and MUAC, a part of somatic growth [43]. Somatic changes vary significantly in age at start and end, speed, and nature, depending on the individual. This finding is in line with research by Nainggolan et al. indicating overweight and obese women, regardless of their MUAC score, had a lower risk of being anemic than those with an average BMI.. The most likely opportunity is the occurrence of anemia in women with low MUAC scores [9]. Nevertheless, several studies have evaluated the relationship between anthropometric indicators and anemia, but this is often inconsistent [9,44]. Further research is needed on the relationship between MUAC and other parts of somatic growth because, in this study, only MUAC showed a significant association with anemia without weight and height.

The strength of this study is to analyze the effect of anemia on the quality of life of adolescent girls. In contrast, in previous studies, this data was minimal, especially in rural areas in Indonesia. In addition, the physical examination, including MUAC, was by the guidelines from UNICEF, and blood samples were taken by professional laboratory personnel and examined by an accredited laboratory. 

The limitation of this study is that there is no examination of thalassemia and feces for the diagnosis of worms. In our recommendation, it is necessary to provide regional laboratories with complete types of blood and stool examinations. In addition, it is very important to explore and increase adolescents’ knowledge, attitude, and practice regarding anemia and its prevention.

## 5. Conclusions

Related factors of anemia in this study were: duration of blood show per menses, iron consumed, weight, height, and MUAC. Anemia in this study was a mild public health problem that influenced the social relationships domain. Duration of blood per menses and MUAC were the most influential factors for anemia. Health education is needed regarding the related factors of anemia and its prevention. Moreover, there should be coordination between stakeholders to overcome the problem of anemia.

## Figures and Tables

**Figure 1 nutrients-14-03777-f001:**
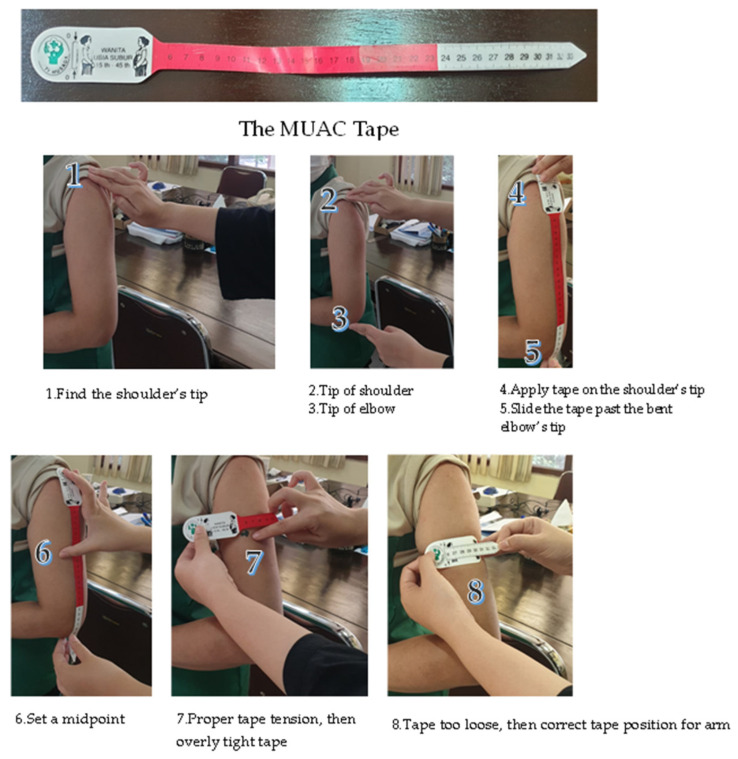
Mid-upper arm circumference (MUAC) Measurement [28].

**Table 1 nutrients-14-03777-t001:** Participants Characteristic.

Characteristic	Non-Anemic n (%)	Anemic n (%)	*p*-Value
Age (median (min-max)) 17 (15–19)	245	41	
Grade			
10	127 (44.4)	20 (7.0)	0.717
11	118 (41.3)	21 (7.3)	
Father’s education			
Primary	73 (25.52)	13 (4.55)	0.620
Secondary	141 (49.30)	25 (8.74)	
University	31 (10.84)	3 (1.05)	
Mother’s education			
Primary	79 (27.62)	14 (4.90)	0.907
Secondary	140 (48.95)	22 (7.69)	
University	26 (9.09)	5 (1.75)	
Father’s occupation			
Employed	6 (2.10)	2 (0.70)	0.634
Not employed	235 (82.17)	38 (13.29)	
Other (Died)	4 (1.40)	1 (0.35)	
Mother’s occupation			
Employed	192 (67.13)	32 (11.19)	0.916
Not employed	52 (18.18)	9 (3.15)	
Other (Died)	1 (0.35)	0 (0.00)	
Living conditions			
Dormitory	27 (9.44)	6 (2.10)	0.503
With family	218 (76.22)	35 (12.24)	
Siblings			
0–2	118 (41.26)	21 (7.34)	0.717
≥3	127 (44.41)	20 (6.99)	
Monthly family income			
<1 million	102 (35.66)	18 (6.29)	0.686
1–3 million	93 (32.52)	17 (5.94)	
≥3–5 million	50 (17.48)	6 (2.10)	
Menarche			
Attained	244 (85.31)	41 (14.34)	0.682
Not Attained	1 (0.35)	0 (0.00)	
Duration of blood flow per each menses			
<5	55 (19.23)	15 (5.24)	0.051
≥5	190 (66.43)	26 (9.09)	
Sanitary pad usage per day			0.983
<3	137 (47.90)	23 (8.04)	
≥3	108 (37.76)	18 (6.29)	
Number of meals per day			
1	30 (10.49)	10 (3.50)	0.156
2	151 (52.80)	19 (6.64)	
3	59 (20.63)	11 (3.85)	
>3	5 (1.75)	1 (0.35)	
Snack consumption			
1	22 (7.69)	7 (2.45)	0.214
2	65 (22.73)	6 (2.10)	
3	51 (17.83)	10 (3.50)	
>3	107 (37.41)	18 (6.29)	
Meat/Fish consumption			
Yes	207 (72.38)	37 (12.94)	0.335
No	38 (13.29)	4 (1.40)	
Fruit/Vegetable consumption			
Yes	223 (77.97)	36 (12.59)	0.515
No	22 (7.69)	5 (1.75)	
Coffee/Tea consumption			
Yes	63 (22.03)	8 (2.80)	0.395
No	182 (63.64)	33 (11.54)	
History of hemoglobin examination			
Yes	30 (10.49)	4 (1.40)	0.649
No	215 (75.17)	37 (12.94)	
History of iron tablet consuming			
Yes	204 (71.33)	30 (10.49)	0.121
No	41 (14.34)	11 (3.85)	
Personal hygiene			
Good	85 (29.72)	16 (5.59)	0.591
Poor	160 (55.94)	25 (8.74)	
Physical activity			
Low	162 (56.64)	30 (10.49)	0.661
Moderate	70 (24.48)	9 (3.15)	
High	13 (4.55)	2 (0.70)	
Height of age (z-score)			
Stunted	88 (30.77)	19 (6.64)	0.202
Not stunted	157 (54.90)	22 (7.69)	
BMI status			
Underweight	48 (16.78)	10 (3.50)	0.556
Normal weight	152 (53.15)	26 (9.09)	
Overweight	45 (15.73)	5 (1.75)	
Anemia classification			
Non anemic	246 (86)		
Mild	13 (4.5)		
Moderate	25 (8.7)		
Severe	2 (0.7)		

**Table 2 nutrients-14-03777-t002:** Nutrient Intake and Quality of Life.

Variable	Non-Anemic	Anemic	Recommendation *	*p*-Value
	Median (Min–Max)	Median (Min–Max)		
Energy	5818.5 (827.13–16,417.72)	6810.41 (97.3–13,831.19)	2125–2250	0.886
Macronutrient intake				
Carbohydrate	597.89 (86.13–1846.23)	600.96 (19.53–1389.23)	292–309	0.866
Protein	293.13 (32.44–754.52).	365.1 (3.31–661.29)	56–69	0.926
Fat	281.79 (38.19–710.44)	318.96 (0.48–640.49)	71–75	0.768
Micronutrient intake				
Vitamin A	5390.78 (44.7–44,742.8)	4903.79 (0.00–16,720.85)	600	0.578
Vitamin B1	2.71 (0.24–8.26)	3.06 (0.01–7.08)	1.1	0.886
Vitamin B2	4.42 (0.36–14.81)	4.41 (0.01–9.94)	1.0–1.1	0.631
Vitamin B6	5.2 (0.59–19.03)	6(0.03–14.7)	1.2–1.3	0.916
Vitamin C	341.07 (0–2048.22)	346.97 (0–2048.22)	65–75	0.861
Vitamin E	28.98 (2.43–145.8)	28.79 (0.0–81.7)	15	0.870
Calcium	2283.27 (44.85–5942.71)	1762.87 (4.83–5942.71)	1000–1200	0.322
Folic acid	717.69 (32.88–2744.72)	634.32 (4.83–2038.71)	400	0.675
Iron	73.87 (2.84–210.97)	59.82 (0.34–210.97)	26	0.490
Zinc	39.65 (4.18–104.53)	43.92 (0.34–91.87)	8–9	0.952
Quality of life				
Physical health	44 (6–75)	44 (19–63)		0.196
Psychological domain	50 (0–94)	50 (19–81)		0.815
Social relationships	50 (0–100)	50 (19–81)		0.024
Environment domain	56 (0–100)	50 (19–75)		0.375

* The nutrient composition of Indonesia reference.

**Table 3 nutrients-14-03777-t003:** Anthropometric and Hematology.

Variable	Non-Anemic	Anemic	*p*-Value
	Median (Min–Max)	Median (Min–Max)	
Weight	49 (33–98)	46 (33–79)	0.048
Height	153 (135–169)	151 (139–160)	0.031
Upper arm circumference	24 (19–33)	23.5 (19–28.5)	0.076
Hematology characteristic			
White blood cells (WBC)	8 (4.60–13.80)	7.3 (4–11.9)	0.009
Lymphocytes	30.7 (15.1–51.20)	31.3 (19.2–60.0)	0.826
Hemoglobin	13.4 (11.4–16.20)	10.3 (7.3–12.7)	<0.001
Red blood cells (erythrocytes)	4.7 (4–6.7)	4.7 (4.1–6.1)	0.751
Hematocrit (HCT)	39.6 (34–45.3)	34 (24.7–37.4)	<0.001
Mean corpuscular volume (MCV)	84.4 (62–92.9)	71.6 (54.5–88.9)	<0.001
Mean corpuscular hemoglobin (MCH)	29 (18.8–32.4)	21.8 (16.6–28.7)	<0.001
Mean corpuscular hemoglobin concentration (MCHC)	34.3 (23.8–36.1)	31 (26.9–34.5)	<0.001

**Table 4 nutrients-14-03777-t004:** The Most Influence Factors of Anemia (Multivariate Analysis).

Variables		Coefficient	*p*	OR (IK95%)
Step 1a	Duration of blood show per menses (1)	0.702	0.058	2.018 (0.977–4.168)
	Iron consume (1)	0.441	0.277	1.555 (0.702–3.445)
	Weight	0.002	0.963	1.002 (0.924–1.087)
	Height	−0.042	0.242	0.959 (0.893–1.029)
	Mid-upper arm circumference (MUAC)	−0.131	0.405	0.878 (0.645–1.193)
	Constant	7.378	0.231	1599.768
Step 2a	Duration of blood show per menses (1)	0.703	0.057	2.020 (0.979–4.168)
	Iron consume (1)	0.443	0.273	1.557 (0.705–3.440)
	Height	−0.041	0.189	0.959 (0.902–1.021)
	Mid-upper arm circumference (MUAC)	−0.124	0.091	0.883 (0.765–1.020)
	Constant	7.195	0.128	1332.62
Step 3a	Duration of blood show per menses (1)	0.737	0.045	2.089 (1.016–4.296)
	Height	−0.045	0.156	0.956 (0.899–1.017)
	Mid-upper arm circumference (MUAC)	−0.126	0.086	0.882 (0.764–1.018)
	Constant	7.849	0.098	4.79
Step 4a	Duration of blood show per menses (1)	0.785	0.032	2.192 (1.072–4.486)
	Mid-upper arm circumference (MUAC)	−0.149	0.036	0.862 (0.750–0.990)
	Constant	1.567	0.354	4.790

## Data Availability

Not applicable.

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
