# Peer review of "Anemia among Adolescent Girls in West Java, Indonesia: Related Factors and Consequences on the Quality of Life"

_nutrients, 2022, doi:10.3390/nu14183777_

Round 1

Reviewer 1 Report

In this study the Authors investigated the factors that cause anemia and the effect of anemia on the quality of life in adolescent girls so that the prevalence of anemia can be prevented from adolescence.

The aim of this study is interesting, however, I have the following comments and questions of the manuscript:

The abstract should be corrected - the second sentence - it is unclear and the conclusions - repetition with description of the results.

Were the consumption data collected at the same time as blood chemistry tests and were there any questions about dietary supplements - this should be added

The conditions for blood collection for tests should also be added in the methods section: time of day, hours, were they the same for all the subjects?

Were the blood tests done immediately or whether the blood (plasma) was frozen - if so, add it in manuscript

There is a lack of important information on the number of girls with and without anemia - table 1, 2 ... and information whether they were of the same age?

I propose to strengthen the sections discussion on the consequences of anemia in girls and the limitation section.

In my opinion, this study has more limitations - was it a representative sample, or was the 24 h dietary recall conducted only once?

Moreover, it was a cross-sectional study, which precludes assessment of the causal relationship between variables. Please explain.

Were there any and what strengths of this study?

In addition, conclusions should correspond to the set goal and be combined with recommendations

Moreover, I would suggest:

- to read the entire work carefully and proofreading.

- to adjust the method of citing references and presenting references  in accordance with the journal guidelines.

Author Response

Dear Reviewer,

We wish to thank Reviewer 1 for the excellent comments and suggestions. We have responded to each of the issues raised. All changes are highlighted using the track change words in the revised manuscript version and in yellow text highlight color in this response letter. (Please see the attachment) 

Best regards,

Reviewer 2 Report

Sari Puspa and other colleagues from Indonesia reported on Quality of Life in anaemic adolescent. Moreover they identified an useful relationship between anemia and the mid upper arm circumference (MUAC). This paper is of interest and in my opinion deserves to be publish

SOME MINOR Improvement are required: 

INTRODUCTION:

- Authors need to mention a differential diagnosis with also haemoglobinopathies and sickle cell disease. Please add two lines on this. Use these References:

Do we need to test blood donors for sickle cell anaemia?A Piccin. Blood Transfusion 8 (3), 137

Sickle cell disease and dental treatment A Piccin, P Fleming, E Eakins, E McGovern, OP Smith, C McMahon. J Ir Dent Assoc 54 (2), 75-79

Insight into the complex pathophysiology of sickle cell anaemia and possible treatment A Piccin, C Murphy, E Eakins, MB Rondinelli, M Daves,  European journal of haematology 102 (4), 319-330   - Needs to report better on the exact relationship between MUAC and BMI   MATERIAL/METHODS - describe and provide a picture on how MUAC is taken - page 3/122 delete respondents - provide list of macro and micronutrients that were measured - page 6/169 check you are repeating "macronutrients" twice   DISCUSSION - page 10/242    spelling incorrect different and difference    

Author Response

Dear Reviewer,

We wish to thank Reviewer 2 for the excellent comments and suggestions. We have responded to each of the issues raised. All changes are highlighted using the track change words in the revised manuscript version and in yellow text highlight color in this response letter. (Please see the attachment)

Best regards,
